# A novel analytical approach for the simultaneous measurement of nitrate and DOC in soil water

Elad Yeshno[1], Ofer Dahan[1], Shoshana Bernstain[1], Shlomi Arnon[2]

[1]Department of Hydrology & Microbiology, Zuckerberg Institute for Water Research, Blaustein Institutes for Desert Research, Ben-Gurion University of the Negev, Israel
[2]Electrical and Computer Engineering Department, Ben-Gurion University of the Negev, Israel

*Correspondence to*: Elad Yeshno (Eladyes@post.bgu.ac.il)

**Abstract.** In this paper, we present a novel approach, enabling the measurement of nitrate concentrations in natural soil porewater containing natural soil dissolved organic carbon (DOC). The method is based on UV absorbance spectroscopy, combined with fluorescence spectroscopy, for simultaneous analysis of DOC and nitrate concentrations. The analytical procedure involves deduction of the absorption caused by the DOC from the total absorbance in the UV range that is attributed to both DOC and nitrate in the water solution. The analytical concept has been successfully tested in soil water samples obtained from five agricultural sites, as well as in water samples obtained from a commercial humus soil mixture. We believe that the new analytical concept can provide a scientific foundation for developing a sensor for real-time nitrate concentration measurements in agricultural soils. As such, it can play a significant role in reducing nitrate pollution in water resources, optimizing input application in agriculture, and decreasing food production costs.

## 1. Introduction

During the last half-century, clear trends of rising nitrate concentrations in groundwater have been observed in aquifers all around the globe (Jin et al., 2012; Kourakos et al., 2012; Liao et al., 2012). The World Health Organization (WHO) has determined that nitrate levels in drinking water should not exceed 50 mg/L (WHO, 2011). At high concentrations, nitrate is especially harmful to infants, where it may cause "blue baby syndrome" (methemoglobinemia) and can lead to severe illness and even death (Lorna Fewtrell, 2004; Thompson et al., 2007). Unfortunately, nitrate contamination is the dominant factor responsible for severe degradation of groundwater and surface resources. On a global scale, the eutrophication and hypoxia of streams, rivers and lake, are mostly attributed to subsurface return flow from nitrate-contaminated groundwater from phreatic aquifers underlying agricultural fields. Moreover, nitrate-contaminated groundwater affects not only terrestrial water resources but marine ecosystems as well. Eutrophication and hypoxia on a large scale have been found in the Gulf of Mexico (Scavia et al., 2003) and the Black Sea (Tolmazin, 1985), as well as severely impacting the Great Barrier Reef, in Australia (Brodie et al., 2012). Overall, nitrate contamination has led to more groundwater disqualification and water well shutdowns than any other contaminant worldwide (Dahan  al., 2014; Osenbrück et al., 2006), with nitrate being considered the most common non-point source groundwater pollutant (Turkeltaub et al., 2016). Numerous studies have linked increased nitrate concentrations in groundwater to the excess use of agricultural fertilizers (Burow et al., 2010; Fisher and Healy, 2008; Kurtzman et al., 2013). As a result, a global regulatory effort to reduce excess fertilizer application to prevent nitrate pollution has recently been undertaken worldwide by environmental protection and water authorities (Bureau, 2018; EPA US and Office of Water, 1994).

Currently, agricultural fertilizer application relies primarily on farmers' experience, expert recommendations, and sporadic soil testing. None of these provide information that is in line with the time scale of N-fertilizer mobilization, consumption, and transformation dynamics in the soil. As such, continuous in-situ nitrate measurement in the soil is essential for optimizing fertilizer application and reducing groundwater pollution potential (Šimůnek and Hopmans, 2009).

Presently, monitoring of soil chemical parameters is performed in water samples that may be obtained using suction cups installed in the soil or through water extraction of soil samples (Kabala et al., 2017; Reck et al., 2019). Water samples collected using these methods need to be transferred to a laboratory for further chemical analyses, or direct analysis on-site using analytical kits (Golicz et al., 2020). Nitrate in the soil is highly soluble, mobile, consumed by plants, and subject to various biochemical transformations. Accordingly, nitrate concentration in the soil may fluctuate dramatically in time scales of hours to days in response to irrigation and precipitation patterns, fertilization, root uptake, and different plant growth phases (Mmolawa and Or, 2000; Turkeltaub et al., 2015). Thus, monitoring nitrate concentrations with conventional tools often does not meet the required time resolution for optimizing fertilizer application and preventing groundwater pollution from excess use. The use of highly sophisticated soil sampling methods can be expected from a dedicated research team but not from farmers who are focused on food production in large scale agricultural setups (Thompson et al., 2007). Therefore, automated, real-time monitoring of soil nitrate concentrations may provide growers with vital information on nutrient availability. Such data is critical for maximized productivity, in addition to water resource protection from nitrate pollution.

Ultraviolet (UV) absorption spectroscopy is one of the most common methods for nitrate analysis in aqueous solutions (Mayerstein and Treinin, 1961; Moorcroft, 2001). Light absorption by nitrate in an aqueous solution takes place at two main wavelength bands on the UV spectrum: (1) the high absorbance band from 200–240 nm and (2) the low absorbance band from 280–340 nm. Tuly et al. (2009) suggested a continuous soil solution monitoring setup to measure nitrate concentration by applying absorbance spectroscopy techniques in a porous

cup installed in the soil. This setup consisted of a porous stainless-steel cup filled with deionized water that was placed in a reservoir of a potassium nitrate solution. Once the solution inside the porous cup achieved chemical equilibrium by diffusion between the cup and the surrounding medium, the absorption spectrum of the solution was measured by a UV dip probe. The dip probe was connected to a spectrophotometer and a UV light source through optical fibers, which enabled continuous measurement of the solution within the suction cup. However,

this method was limited by two main factors: (1) achieving chemical equilibrium between the porous cup and the surrounding medium is a slow process, especially in unsaturated sediment that has limited water storage. The time lag between the actual variation in soil nitrate concentration and its measurement in the porous cap precludes the real-time measurement of the rapid changes in the soil nitrate concentration, and (2) the presence of natural soil dissolved organic carbon (DOC) limits the UV absorption spectroscopy analysis since both nitrate and soil DOC absorb UV light in a similar wavelength range (Causse et al., 2017; Ferree and Shannon, 2001; Shaw et al., 2014).

DOC interference to the nitrate absorbance spectrum can be reduced by applying a combination of the UV-VIS absorption spectrum and multivariant statistical models such as Partial Least Square Regression (PLSR) (Avagyan et al., 2014; Etheridge et al., 2014; Rieger et al., 2006). PLSR is similar to Principal Component Regression (PCR), when dealing with multidimensional data sets, however, PLSR has the additional advantage

of maximizing the correlation between the variables and a parameter of interest (Wehrens, 2011). Vaughan et al., (2017) had demonstrated the use of PLSR based method along with in-situ optical apparatus to estimate nitrate concentration in watersheds of urban, agricultural, and forested lands. However, the general DOC and nitrogen concentration in their research had ranged between 0-10 ppm, one order of magnitude less than the nitrate and DOC levels found in the shallow subsurface porewater of agricultural soils (Dahan et al., 2014; Yeshno et al.,

2019).

          Recently, a monitoring setup for real-time soil nitrate concentration measurement was developed, which is based on continuous spectral analysis of soil porewater (Yeshno et al., 2019, and 11-158 PCT, 2018) in an optical flow cell that is connected to a specially designed porous cup (suction lysimeter). This enables the creation of a continuous low rate flux of soil porewater flowing from the soil through the cell. The absorption spectrum of

the soil porewater in the optical flow cell is continuously recorded and analyzed to achieve real-time nitrate concentration measurements. The analysis involves an algorithm that scans the entire absorption spectrum of the soil porewater to identify an optimal wavelength where DOC interference with nitrate measurement is minimal. Use of the method showed that the optimal wavelength is site-specific, and the calibration equations for nitrate concentration are consistent over a long-time duration. Measurements conducted for two years in four different

agricultural fields and through a series of column experiments offered detailed information on time variations in soil nitrate concentrations (Yeshno et al., 2019). Despite this significant breakthrough in continuous soil nitrate concentration measurement, the technology is still too cumbersome and impractical for commercial agricultural applications since it is based on a wide-band UV spectrophotometer and a bulky deuterium lamp as a UV light source.

Reducing the cost, size, and energy consumption of a spectroscopy-based apparatus for measuring soil porewater nitrate concentrations requires a shift in applied technology from the commonly used UV deuterium lamp to a less expensive LED-based UV light source. In addition, the commonly used wide spectrum spectrophotometer can be replaced with a more affordable photodetector. However, currently, most affordable, commercially available UV-LEDs do not possess the range of the high nitrate absorbance band (200–240 nm),

but only of the low nitrate absorbance band in the range of $300 \pm 20$ nm. Nevertheless, the 300-nm spectral band can still be used for the nitrate absorption spectral analysis (Moo et al., 2016). However, this band is also sensitive to DOC presence. Moreover, DOC's impact on the absorption at 300 nm is inconsistent and depends on the soil water's specific chemical composition (Yeshno et al., 2019). Thus, directly analyzing nitrate concentration from the absorbance in the wavelengths surrounding the 300-nm band in water samples originating from agricultural

soils is challenging.

        In this paper, we present a novel approach, enabling nitrate estimation in porewater samples containing DOC with a variety of concentrations and chemical compositions. The analysis is based on UV absorbance spectroscopy in the vicinity of 300 nm combined with simultaneous fluorescence spectroscopy with excitation/emission at 350/451 nm for characterizing DOC impact on the absorption pattern. The independent

DOC measurement allows subtracting DOC's contribution to the adsorption at 300 nm ((nitrate + DOC) – DOC = nitrate), thus enabling the estimation of nitrate concentration. The analytical procedure involves the spectral analysis of a matrix of soil porewater solutions containing variable combinations of DOC and nitrate concentrations from different agricultural soils. The resulting absorption and fluorescence database was used to develop a polynomial calibration equation that calculates nitrate concentration as a function of the absorbance at

300 nm and the DOC concentration (patent pending: PCT/IL2020/050645). The rationale behind the selection of these wavelengths for the analysis is related to the commercial availability of LED light sources in these wavelengths. Obviously, developing a robust soil nitrate sensor that is commercially affordable for widespread use in agricultural applications will lead to the optimization of fertilizer application and the reduction of water resource pollution.

**2. Materials and methods**

**2.1 Study sites and soil samples**

        Soil samples from five different agricultural fields, located on the coastal plain of Israel, were collected and analyzed for this study. The samples were collected from organic and conventional greenhouses that grow vegetable crops, an open field that is used for rotating mixed crops, and a citrus orchard (Table 1). These sites

were chosen to represent a spectrum of typical agricultural practices in different soils. In addition, a commercial humus soil mixture (Dovrat. LTD, Israel) was also examined to represent the potential impact of DOC originating from compost enrichment on a soil porewater spectral analysis. These sites have been intensively studied in the past, and additional information can be found in previous publications (Dahan et al., 2014; Turkeltaub et al., 2014, 2015, 2016) and in the Supporting Information section (Section S1).

**2.2 Soil water samples preparation and chemical and spectral analyses:**

        Soil water extract's samples were obtained by creating a mixture of the different soils with double distilled water (DDW). The soil extracts were left to stand for 24 h in order to achieve chemical equilibrium of

the soil DOC. The soil and liquid phases were separated by a standard laboratory centrifuge, and the suspended solids were removed by a 0.22-µm membrane filter.

The initial values of DOC and total nitrogen (TN) in the sample were estimated by an Analytic Jena multi N/C 2100s TOC/TN analyzer, while nitrate concentration was determined by a Dionex ICS 5000 Ion chromatograph. The absorption of the samples at 300 nm was determined using a TECAN Spark 10M multimode microplate reader spectrophotometer. The light absorbance was defined by the Lambert-Beer equation. (Equation 1):

$$Absorbance = -log_{10}\frac{I}{I_0} \tag{1}$$

where $I$ is the light intensity after passing through the examined solution, and $I_0$ is the light intensity after passing through a reference water sample (DDW).

A fluorescence spectroscopy technique was used to measure the DOC concentration in the examined solution using a TECAN Infinite M200 spectrophotometer with excitation (EX)/emission (EM) at 350/451 nm

(Mostofa et al., 2007; Painter et al., 2018; Parlanti et al., 2000). An important advantage of DOC fluorescence spectroscopy is that it is not affected by the presence of nitrate in the solution, as only DOC has the chemical structural complexity which comprises the aromatic rings required to have fluorescence characteristics at the UV range. Moreover, when DOC is targeted as the main chemical component for chemical analysis, UV fluorescence spectroscopy is commonly applied if the samples containing dissolved nitrate or iron instead of absorbance

spectroscopy techniques (Bridgeman et al., 2011).

The chemical and optical data of the soil water solution was analyzed by the MATALB 2019b curve fitting tool to obtain the polynomial equation for nitrate estimation, p-value, correlation coefficient ($R^2$) and RMSE values.

## 3. Results and discussion

### 3.1 Absorption spectroscopy for nitrate estimation in soil water containing DOC

The application of UV absorption techniques to aqueous nitrate solutions commonly results in a linear correlation between the absorption rate and the nitrate concentration (Edwards et al., 2001; Ferree and Shannon, 2001; Michael et al., 2017; Moo et al., 2016; Tuli et al., 2009). However, this correlation is not straightforward in natural soil water containing DOC, as UV absorption by the organic matter, in conjunction with the nitrate absorption, leads to an increase in the overall absorption in the examined solution. For example, a series of solutions with variable nitrate concentrations and a relatively constant DOC concentration shows a linear correlation with the absorption values (curve a in Figure 1). However, a similar series of nitrate concentrations, with an increased DOC level, shows a linear correlation as well, yet with a greater absorbance level (curve b in Figure 1). Nevertheless, the similarities between the trendline associated with the low DOC levels and the trendline associated with the high DOC levels imply that the DOC contribution to the overall absorbance is consistent and quantifiable.

However, examining the relation between nitrate and DOC concentrations and the UV absorption of a large solution matrix may not be as intuitive as the 2D model presented in Figure 1. In order to assess the specific contribution of nitrate and DOC to the total absorption at 300 nm a matrix of solution containing varying concentration of DOC and nitrate was made from the soil water extract for each study site. The solution matrix was obtained through the preparation of a series of replicates of different DOC concentrations for each soil water extract. Each replicate was spiked with a variable volume of 10,000 mg/L standard potassium nitrate solution, to achieve 4–6 different nitrate concentrations per DOC level. As a result, a matrix composed of 25–30 samples with variable combinations of DOC and nitrate concentrations, ranging from zero to ~1000 mg/L nitrate and zero to ~100 mg/L DOC was produced from each agricultural site (Table 2). Additional databases for the remaining five soil type samples can be found under section S2 in the Supporting Information.

The data from the three variables (DOC/nitrate concentration and absorption values) were projected on a 3D domain (Figure 2). The data distribution in space, which is geometrically defined as a plane, was mathematically quantified by applying a multivariate regression model. From this model, the nitrate concentration in the sample could be estimated as a function of the DOC concentration and absorbance at 300 nm (Equation 2):

$$Nitrate(DOC, Abs) = P_{00} + P_{01} \times Abs + P_{10} \times DOC \qquad (2)$$

where *DOC* indicates the DOC concentration (in mg/L), *Abs* indicates the absorbance as measured at 300 nm (a.u.), and $P_{00}$, $P_{10}$, and $P_{01}$ are the coefficients as obtained from the regression model.

Yet, the contribution of the DOC to the overall absorbance can differ from site to site, due to variations in DOC chemical and optical characteristics in the different soils. The composition of the organic matter that constitutes the DOC is affected by site-specific characteristics such as: type of agricultural crop, biological activity in the soil, and the type of compost and fertilizer used at the site (Kalbitz et al., 2000; Nelson et al., 1992; Tian et al., 2010). Therefore, the polynomial calibration equation for nitrate must be site-specific. Nonetheless, it was concluded that the impact of DOC on the absorption spectrum, due to its chemical composition, remained relatively constant through time and is a site-specific feature (Yeshno et al., 2019). This implies that once an initial

calibration equation is obtained, it can be used for that particular site repeatedly in a long duration, making real-time continuous measurement of soil nitrate concentration feasible.

Further validation of the concept was performed by reproducing similar multivariate calibration models and their calibration coefficients for estimating nitrate in the soil water of five additional agricultural sites (Table 3). The polynomial calibration equations were obtained for matrices of water samples containing variable DOC concentrations, ranging from 1.6 to about 200 mg/L, and nitrate concentrations ranging from a few mg/L to about 1000 mg/L. Nitrate concentrations predicted by the multivariate model had a correlation of $R^2 > 0.93$ and an RMSE ranging from 39.2 to 66.4 mg/L nitrate with the observed nitrate concentration in the solution (Figure 3). Detailed information on the range of concentrations and the 3D projected data and quality of the fitted data can be found at Table 4 and under section S2 in the Supporting Information section.

**3.2 DOC analyses using fluorescence spectroscopy**

In order to perform real-time analyses of nitrate in the water samples using the presented concept, the DOC concentration in the examined solution needs to be continuously substituted in the polynomial calibration equation (Equation 2), along with the absorption at 300 nm. Thus, an independent and simultaneous analysis of DOC, along with the absorbance measurement, needs to be performed to determine the DOC concentration in the examined solution.

One way of providing an independent, simultaneous measurement of DOC in the soil solution is by fluorescence spectroscopy. Fluorescence emission values of the soil water samples at 451 nm, which had known DOC concentrations, were used to develop a calibration curve for each site (Figure 4). The relation between DOC concentration and fluorescence intensity in the water samples from the different fields shows that each field has its own unique calibration equation due to the variation in the DOC chemical characteristics. As previously mentioned, the composition of the organic matter that constitutes the DOC and its chemical characteristics are site-specific properties. Therefore, similarly to the polynomial of the nitrate calibration equation, the calibration curves for the DOC concentrations are also required to be site-specific.

**3.3 Limitation in applications of UV-LEDs for nitrate analyses by absorption spectroscopy**

As previously mentioned in this paper, UV absorption by aqueous nitrate can be divided into two distinct sections: (1) a higher absorbance band around 220 nm and (2) a lower absorbance band at 300 nm. While both absorbance bands are known to have direct correlations with nitrate concentration, the absorbance at 220 nm absorbs light at two orders of magnitude higher than the absorbance at 300 nm. Consequently, the 300 nm absorbance, which stretches at values from 0–1 (a.u.), corresponding to the relevant nitrate and DOC concentrations in this research, is characterized by a low signal (nitrate absorbance) to noise (DOC absorbance) ratio. Therefore, nitrate measurement at 300 nm is naturally more vulnerable to measurement errors, reducing the measurement accuracy, especially at the lower concentrations (< 100 mg/L). With the purpose of assessing nitrate measurement in the higher absorbance band, we have tested the calibration approach for the humus soil mixture water samples, through the absorbance measured at 235 nm by a standard laboratory spectrometer (Figure 5). The calibration process that for the higher absorbance band of 235 (Figure 5) was practically identical to the one that was produced for the 300 nm (Figure 3e). A significant increase in the quality of the nitrate analysis can be seen when calibration was carried at 235 nm, and consequently, once such LED will become commercially available,

235 a great improvement can be obtained to the accuracy of the sensor. However, with the currently limited availability of deep UV LEDs on the market, the 300-nm LEDs remain the most affordable and attainable candidate for a UV light source for developing a nitrate monitoring sensor.

## 4. Summary and Conclusions

In the framework of this research, we have developed a new concept that enables nitrate estimation in
240 soil water using UV absorption spectroscopy. The new concept allows the use of LEDs as a light source when performing spectral analyses, which are more affordable, compact, and power-efficient than the commonly used deuterium UV light source. Moreover, the newly developed concept accounts for DOC measurement interference, which is a known issue when applying UV absorption spectroscopy techniques to nitrate analyses in soil water. Due to the currently limited availability of UV-LED light sources in the deep UV range, the concept was
245 developed and validated at a wavelength of ~300 nm.

The simultaneous analytical procedure for DOC and nitrate was tested on soil water extracted from five agricultural soils and from a soil mixture of commercial agricultural compost. The results indicate a reasonable correlation between the predicted and observed values of a series of samples obtained for each soil, with variable nitrate and DOC concentrations. Nevertheless, the non-ideal conditions when applying the new concept at 300
250 nm led to some measurement errors, which are not in line with a state-of-the-art standard laboratory apparatus. However, the concept is quite suitable for the practical demands of an onsite and real-time nitrate monitoring sensor in agricultural and environmental applications. We hope that this work will establish the groundwork for the development of an affordable LED-based, electro-optical nitrate monitoring sensor. Such a system, which does not rely on the standard cumbersome light sources, is expected to be efficient in power consumption, size,
255 and production costs and thus suitable for the requirements of both farmers and researchers. We believe that this nitrate and DOC monitoring concept may lead to a higher level of precision farming and the sustainable use of associated water sources while optimizing yield and decreasing food production costs.

**Authors contribution:**

Elad Yeshno (Ph.D. student) - Conducting the experiment, data analysis, main writer. Shoshana Bernstein (M.Sc. student) - laboratory analysis, conducting experiments, writing. Ofer Dahan (project PI) - designing the experimental setups, data analysis, writing. Shlomi Arnon (project PI) - formulation of the analytical concept, designing the experimental setup, data analysis, writing.

**Acknowledgments**

         The authors wish to express their great appreciation to Michael Kugel, who assisted with each and every technical aspect of the project while providing outstanding advice for the laboratory and field experiments.

**Financial support**

This research has been supported by the KAMIN Framework (Israel Innovation Authority, grant no. 67854), and the Israel Ministry of Agriculture and Rural Development (Eugene Kandel Knowledge Centers) as part of the program The Root of the Matter: The root zone knowledge center for leveraging modern agriculture.

**Competing interests:**

The author of this text declares that neither him nor their co-authors have any competing interests.

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

**Tables and Figures:**

**Table 1: Sampled field sites location and soil type.**

| Site location | Site type | Soil type |
|---|---|---|
| Afek | Open crop field - inland | Clay soil |
| Zikim | Conventional greenhouse | Sandy loam |
| Zikim | Organic greenhouse | Loam |
| Nir Galim | Citrus orchard | Loam |
| Nir Galim | Open crop field - coastal | Sandy loam |
| N/A | N/A | "Dovrat" Commercial compost soil mixture (sandy loam) |

**Table 2: Database obtained by analysing the water samples extracted from the conventional greenhouse soil.**

| Sample ID | Concentration (mg/L) | | Absorption at 300 nm (a.u.) | Sample ID | Concentration (mg/L) | | Absorption at 300 nm (a.u.) |
|---|---|---|---|---|---|---|---|
| | DOC | Nitrate | | | DOC | Nitrate | |
| 1 | 213±0.2 | 73±4 | 1.4444 | 15 | 26.6 | 9±1 | 0.2158 |
| 2 | 209±0.2 | 267±16 | 1.443 | 16 | 26.1 | 205±12 | 0.2237 |
| 3 | 203±0.2 | 546±33 | 1.4411 | 17 | 25.3 | 485±29 | 0.2406 |
| 4 | 199±0.2 | 722±44 | 1.4175 | 18 | 24.9 | 663±40 | 0.2556 |
| 5 | 193±0.2 | 975±59 | 1.4117 | 19 | 24.2 | 917±56 | 0.2757 |
| 6 | 106±0.1 | 36±2 | 0.7401 | 20 | 13.3 | 5 | 0.1263 |
| 7 | 104±0.1 | 232±14 | 0.7757 | 21 | 13 | 201±12 | 0.1452 |
| 8 | 101±0.1 | 511±31 | 0.7729 | 22 | 12.7 | 481±29 | 0.1561 |
| 9 | 99±0.1 | 688±42 | 0.7589 | 23 | 12.4 | 658±40 | 0.1719 |
| 10 | 96.7±0.1 | 942±57 | 0.7651 | 24 | 12.1 | 913±55 | 0.1838 |
| 11 | 53.2±0.1 | 18±1 | 0.3893 | 25 | 6.65 | 2 | 0.0837 |
| 12 | 52.2±0.1 | 214±13 | 0.3963 | 26 | 6.52 | 198±12 | 0.0947 |
| 13 | 50.7±0.1 | 494±30 | 0.4082 | 27 | 6.33 | 478±29 | 0.1153 |
| 14 | 49.7±0.1 | 671±41 | 0.433 | 28 | 6.22 | 656±40 | 0.1275 |
| 15 | 48.4±0.1 | 926±56 | 0.4472 | 29 | 6.05 | 911±55 | 0.1503 |

**Table 3: Calibration coefficients for the multivariate regression model of the study site's obtained water samples.**

|  | $p_{00}$ | $p_{01}$ | $p_{10}$ |
|---|---|---|---|
| Conventional greenhouse | -338 | 10800 | -71.8 |
| Humus soil mixture | -375.3 | 11270 | -124.2 |
| Organic greenhouse | -229.8 | 8304 | -68.47 |
| Open crop field - coastal | -378.2 | 11620 | -235.9 |
| Citrus orchard | -328.8 | 9463 | -20.71 |
| Open crop field - inland | -395.1 | 12940 | -46.96 |

**Table 4: Nitrate predication statistics and DOC/Nitrate concentration range for the examined soil water.**

|  | $R^2$ | RMSE | P-values | Concentrations range (mg/L) | |
|---|---|---|---|---|---|
|  |  |  |  | DOC (Error ±0.1%) | Nitrate (Error ±6%) |
| Conventional greenhouse | 0.964 | 66.4 | 2.69E-20 | 6.05 - 213 | 2.27 - 975.3 |
| Humus soil mixture | 0.9589 | 39.21 | 9.67E-17 | 6.8 - 20 | 18.5 - 525.2 |
| Organic greenhouse | 0.9335 | 57.54 | 8.07E-17 | 4.8 - 50 | 2.6 - 619.4 |
| Open crop field - coastal | 0.9693 | 59.82 | 2.27E-17 | 3.3 - 59.7 | 5.1 - 983.3 |
| Citrus orchard | 0.9575 | 47.68 | 3.96E-17 | 9.5 - 99.8 | 2 - 618.4 |
| Open crop field - inland | 0.9771 | 51.97 | 2.26E-20 | 1.63 - 28.65 | 3.2 - 955.8 |

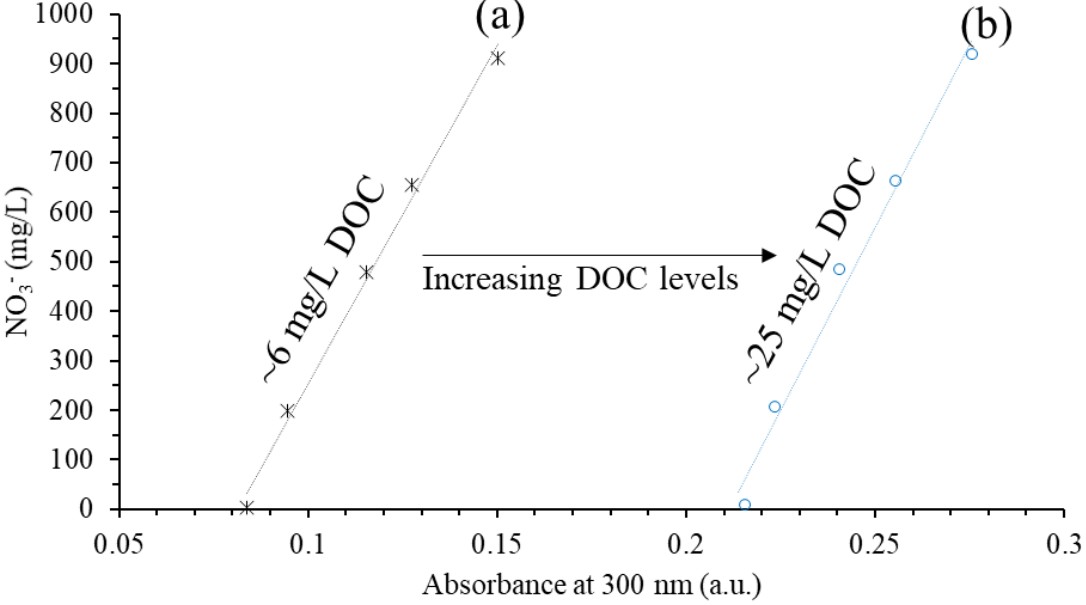

**Figure 1 - Nitrate concentration vs. absorption at 300 nm of soil water samples from the conventional greenhouse.**

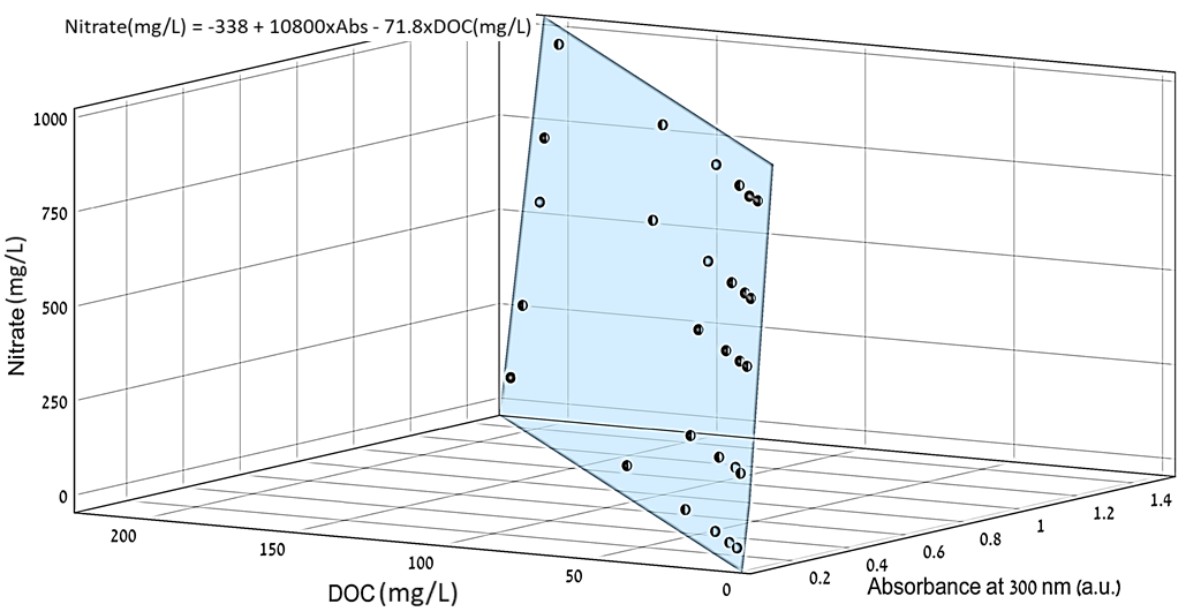

**Figure 2 - 3D projection of the nitrate concentration as a function of the DOC concentration and absorbance at 300 nm for water samples obtained from the conventional greenhouse.**

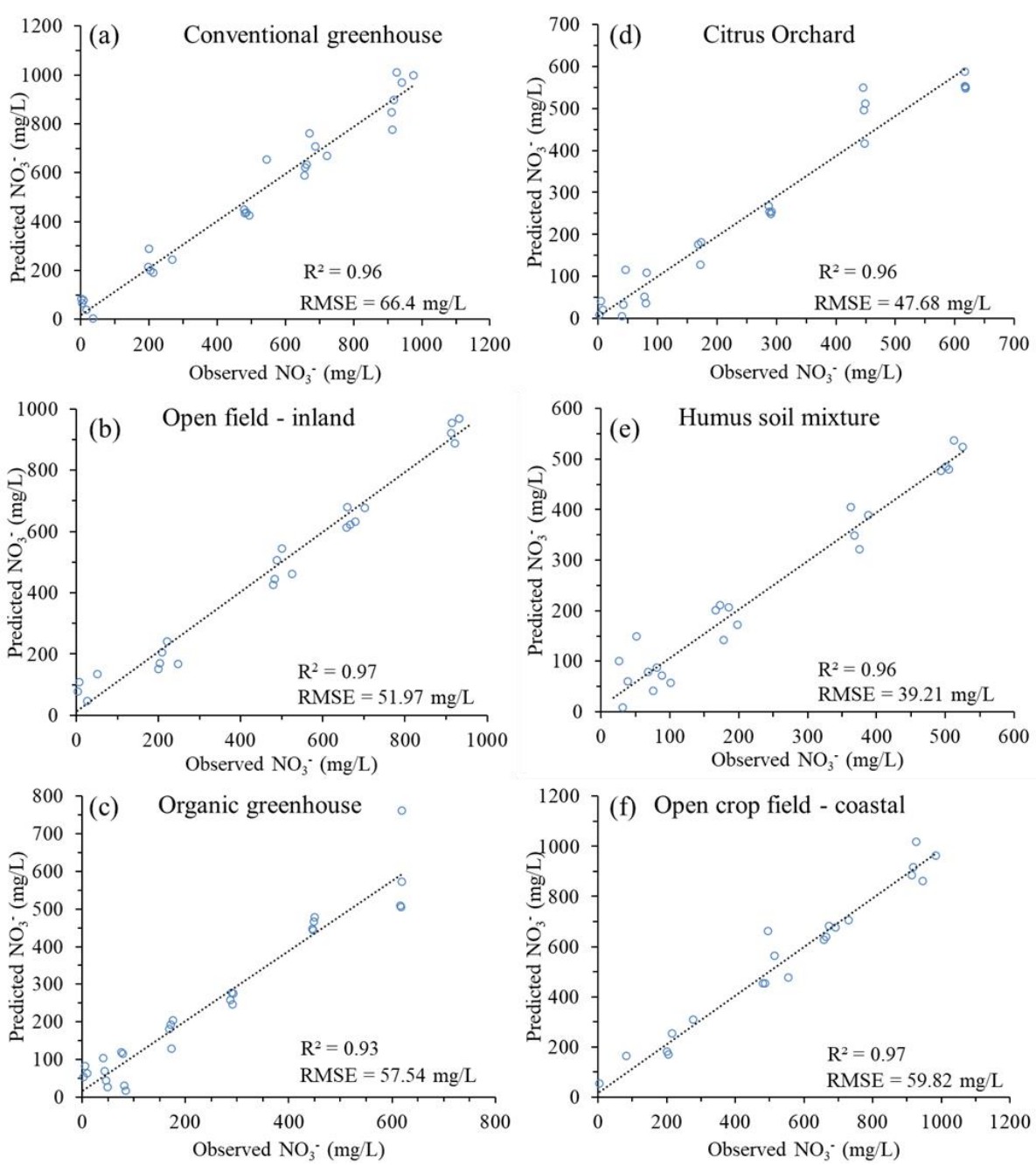

**Figure 3 - Observed vs. predicted nitrate concentrations.**

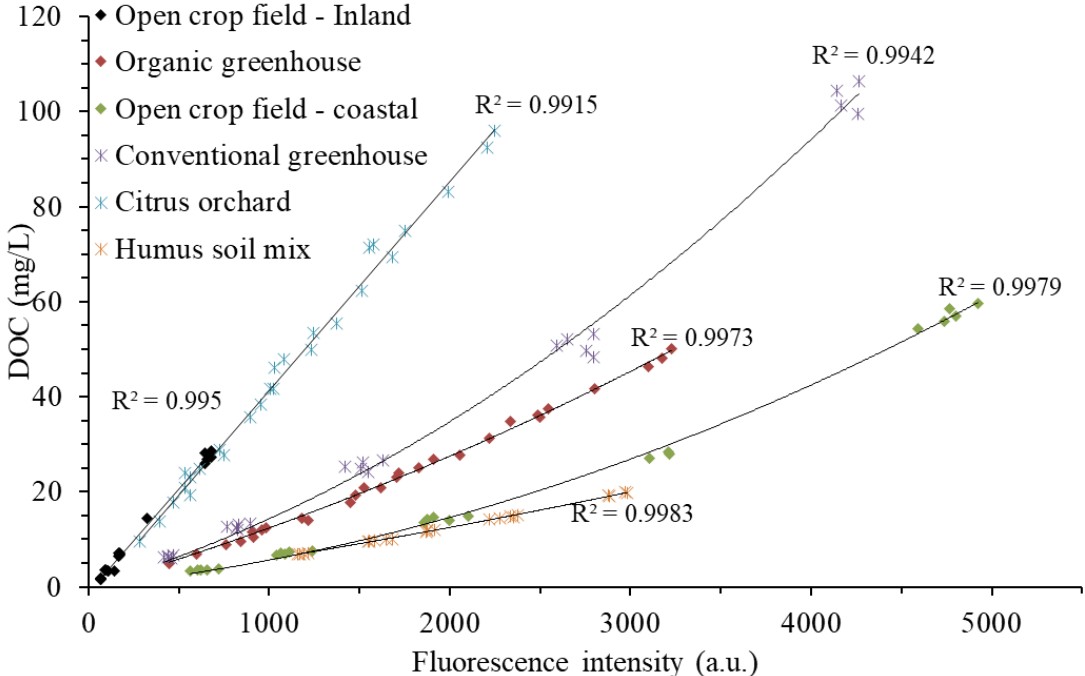

**Figure 4 - DOC concentration vs. fluorescence intensity at 451 nm.**

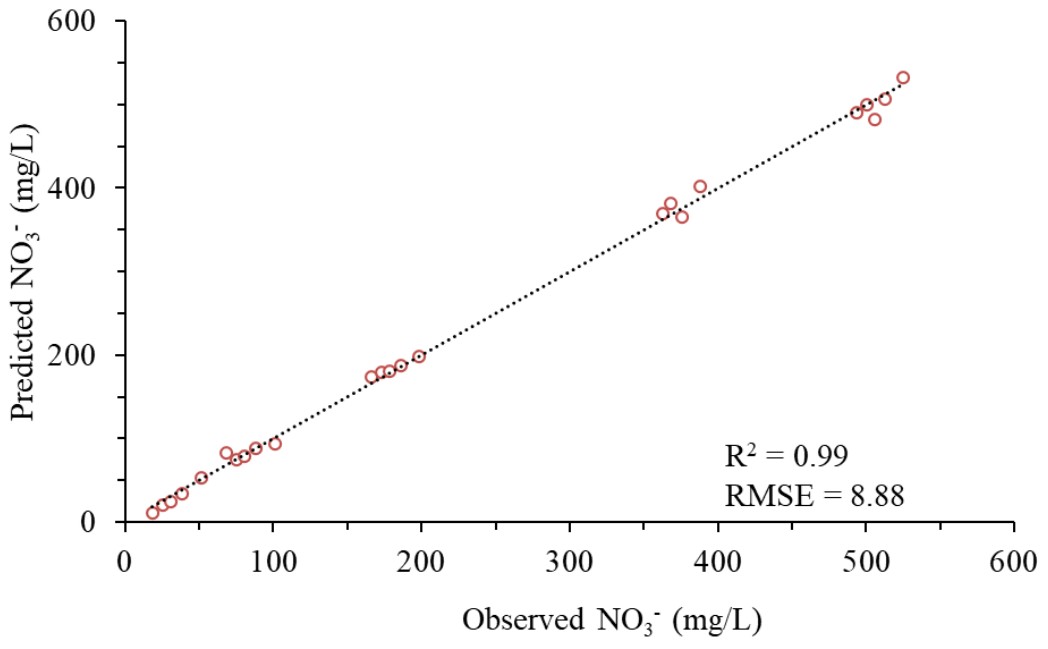

**Figure 5 - Observed vs. predicted nitrate concentrations, for the humus soil mixture water extract, when the calibration equation was obtained for absorption at 235 nm using a standard laboratory spectrometer.**