# Peer review of "A novel analytical approach for the simultaneous measurement of nitrate and DOC in soil water"

_Hydrology and Earth System Sciences, 2020_

## Referee Comment (RC1) · Anonymous Referee #1 · 2 Oct 2020

**General Comments**

This is a very well written and well done study that should be of broad international interest. The methods were clear and the Results and Discussion nicely done on the whole. I believe that this paper is suitable for publication after minor revision.

**Specific Comments**

1) Concentrations of nitrate and DOC are expressed as ppm, which is not a SI unit. My preference would be to change concentrations to units of mg/l throughout the paper (including tables and figures) and supplementary material. Such units are also better for flux calculation in case one were so inclined.

2) In the Methods, I wonder if iron caused any interference in the UV absorption measurements. If not, please state as such. If so, please explain how this was handled.

3) In the Methods, I also think that a photograph or diagram of the experimental set-up would be useful for readers. Please add.

4) I understand that this paper is about soilwater. However, it struck me as I was reading this paper, if it was worth mentioning in the Introduction and/or Discussion how such techniques have been used to measure DOC and nitrate concentrations in streamwater. Otherwise, one might not know that such techniques have been used elsewhere in hydrology. I think it is important to provide the reader with some context on other measurements of DOC and nitrate for sake of completeness and context. I am only asking for a nod to such work. A paper by Vaughan et al (2017) shows streamflow DOC and nitrate concentrations in streamwater for forested, agricultural, and urban watershed (see: https://agupubs.onlinelibrary.wiley.com/doi/full/10.1002/2017WR020491). Such a paper might be worth mentioning with regard to the above but there are many other options.

---

## Referee Comment (RC2) · Anonymous Referee #2 · 6 Oct 2020

General comments This paper presents an advance in the analysis of soil water. Maybe not a giant leap but surely an interesting technique. The text is understandable and fluent and the reader is able to get a fairly complete idea of the work done. However, some important points should be discussed. How easy is it for the farmer to prepare samples for analysis in a reproducible way? No explanation is given on how the estimation of figure 5 has been obtained.

Specific comments The authors state that "the polynomial calibration equation for nitrate must be site-specific.". This can be a limitation to a system that should be used by "farmers who are focused on food production in large scale agricultural setups".

Five soil samples can be few to represent all possible real situations. What is the accuracy of the value of Table 1? The authors state that "An important advantage of DOC fluorescence spectroscopy is that it is not affected by the presence of nitrate in the solution." Sure? Why? It would have been better a version of Figure 3 where the nitrate prediction is based on the DOC measurement made with the fluorescence technique. Please explain why only the points of figure 3b are vertically aligned.

Technical corrections The correct spelling is "humus" and not "hummus". Line 168, "Figure 3" is again "Figure 2". Line 172, "x" should be the multiplication sign.

---

## Referee Comment (RC3) · Anonymous Referee #3 · 20 Oct 2020

The Manuscript "A novel analytical approach for the simultaneous measurement of nitrate and DOC in soil water" by Elad Yeshno et al. presents a measurement setup which tests a combination of two analytical tools to demonstrate the potential of wavelength specific detection of soil water nitrate concentrations in carbon rich soils. The authors applied a combination of UV absorption spectrometry (around 300 nm) with fluorescence spectrometry (excitation/emission at 350/451 nm) to demonstrate the possible feasibility of this setup when applied with single banded LED technology. The study seems to be embedded in a research project or development program, which delivered already an interesting publication with regard to a possible application of the method (Yeshna et al. 2019, HESS (within the reference list this citation misses the journal

name)).

The Authors present in a very straightforward manner what they have done due to which reasons and how the outcomes support the choice of the combined measurement techniques presented in this paper.

However, I have some points that need to be discussed prior to the editor′s decision about a possible publication in HESS:

- Generally, as a reader I would really appreciate the demonstration of a (-n inexpensive) LED-based technology which is able to directly measure in-situ soil water concentrations of DOC and nitrate. But it needs to be discussed whether a feasibility study for the application of analytical tools fits into the scope of HESS or better into another journal format (e.g. the EGU Journal "Geoscientific Instrumentation, Methods and Data Systems").

- The presentation of the study does not follow the typical/expected structure and misses some details:

While the introduction section describes the background of the research problem and the linkage between DOC and Nitrate UV-absorption, it misses a review of other methods applicable to the problem presented in the paper. The interference of UV-absorption of nitrate and DOC is known and several studies (some of them are cited in the paper, but in a different context) use two different wavelengths within the UV-spectrum in combination with statistical models to overcome this problem. Even several commercial UV-Vis spectrometers for a parallel direct measurement of DOC and nitrate in aqueous solution are available on the market and widely in use e.g. in monitoring stations in wastewater treatment plants. The principle of this approach is at least similar to the one applied within this study and could be transferred to an application with LED-technology, likewise. The authors should explain how their approach improves the current state of the art technology (in their approach a second analytical tool is required).

The description of the methodology regarding sampling and analytical procedures is very clear and to the point. The method development, e.g. the derivation of the 2D-Model, which is then applied to the data, is presented within the result & discussion section. Here I would expect a clear difference between method description and acquired parameter values. As mentioned by reviewer 2, I would have expected a larger data set, covering a broader range of possible DOC concentrations than 6 and 25 ppm DOC, respectively.

The specific results and the quality of the presented method are not presented in detail. In addition to the figures, there is only an overall description of the correlations between predicted and measured nitrate concentrations as well as the total range of RMSE. Since this is a new methodological setup, I would have expected at least a table, where the quality differences for the different field sites with regard to the differences in background DOC concentrations is presented to the reader.

The discussion section does not compare the acquired results to other studies in the field (the authors present not a single citation of other literature from the field within this section). Since the presented method needs a site-specific calibration (comparable to the UV-vis-approach), the advantage of this method over others is not clear to me, e. g. whether this setup provides a higher accuracy than UV-vis based spectrometers.

Overall, I recommend major revisions.

---

## Author Comment (AC1) · 3 Dec 2020

General Comments: This is a very well written and well done study that should be of broad international interest. The methods were clear and the Results and Discussion nicely done on the whole. I believe that this paper is suitable for publication after minor revision.

[Figure]

Reply to general comments: We would like to thank the reviewer for the encouraging words, both on the importance of the research and the quality of the reported article.

Specific Comments: Comment 1: Concentrations of nitrate and DOC are expressed as ppm, which is not a SI unit. My preference would be to change concentrations to units of mg/l throughout the paper (including tables and figures) and supplementary material. Such units are also better for flux calculation in case one was so inclined. Reply to specific comment 1: We accept the reviewer comment and have revised the manuscript, figures and supporting information files accordingly.

Comment 2: In the Methods, I wonder if iron caused any interference in the UV absorption measurements. If not, please state as such. If so, please explain how this was handled. Reply to specific comment 2: It is indeed correct that dissolved iron absorbs UV light at a similar wavelengths region to nitrate and can potentially cause interference when applying UV absorption spectroscopy techniques. However, iron has very low absorbance around the examined area of the UV spectrum in this research (300 nm), even at relatively high concentrations. For example, Shaw et al., (2014) showed that $Fe^{2+}$ at a concentration of 250 mg/L has an absorbance intensity of less than 0.1. additionally, during their research, Shaw et al., (2014) had analyzed the chemical composition of water samples obtained at 27 different sites, and found that $Fe^{2+}$ was exhibit in the samples at very low concentrations (<0.29 mg/L). We therefore deduced that iron interference on the UV absorption spectrum at the sample analyzed in this research would be negligible.

Comment 3: In the Methods, I also think that a photograph or diagram of the experimental set-up would be useful for readers. Please add. Reply to comment 3: The research presented in this article is merely a proof of concept and was therefore conducted solely on benchmark laboratory apparatus. The new concept can indeed perform as the analytical basis onto which a LED-based nitrate sensor can be developed. However, a LED-based analytical system was not yet developed in the framework of this research.
Comment 4: I understand that this paper is about soilwater. However, it struck me as I was reading this paper, if it was worth mentioning in the Introduction and/or Discussion how such techniques have been used to measure DOC and nitrate concentrations in streamwater. Otherwise, one might not know that such techniques have been used elsewhere in hydrology. I think it is important to provide the reader with some context on other measurements of DOC and nitrate for sake of completeness and context. I am only asking for a nod to such work. A paper by Vaughan et al (2017) shows streamflow DOC and nitrate concentrations in streamwater for forested, agricultural, and urban watershed (see: https://agupubs.onlinelibrary.wiley.com/doi/full/10.1002/2017WR020491). Such a paper might be worth mentioning with regard to the above but there are many other options.

Reply to comment 4: We accept the reviewer's comment and revise the manuscript accordingly to add more extensive and clarifying details on additional techniques for aqueous nitrate and DOC estimation (lines 76-85). Indeed, past research has already shown the ability to measure nitrate and DOC at surface and stream water using spectral methods. However, the work presented by Vaughan et al (2017) and additional references cited in their paper shows calibration methods which are based on Partial Least Squares Regression (PLSR) (Avagyan et al., 2014; Etheridge et al., 2014; Rieger et al., 2006). PLSR has shown excellent results predicting both nitrate and DOC using absorbance spectroscopy at the UV-VIS range. Yet the method presented in their research required the absorbance data at a broad spectrum on the UV-VIS to obtain a calibration and perform turbidity noise reduction (220-750nm). However, the newly developed concept presented in this paper focuses on a robust method that would merely require a single wavelength for each chemical component (DOC/nitrate) so that the method could be used as the base for an affordable LED-Based sensor for agricultural soils. Moreover, by applying further engineering know-how, a practical optical apparatus can be developed to utilize the same methods presented in this research on surface water or streamwater as well.

References: Avagyan, A., Runkle, B. R. K. and Kutzbach, L.: Application of high-resolution spectral absorbance measurements to determine dissolved organic carbon concentration in remote areas, J. Hydrol., 517, 435–446, doi:10.1016/j.jhydrol.2014.05.060, 2014.

Etheridge, J. R., Birgand, F., Osborne, J. A., Osburn, C. L., Burchell, M. R. and Irving, J.: Using in situ ultraviolet-visual spectroscopy to measure nitrogen, carbon, phosphorus, and suspended solids concentrations at a high frequency in a brackish tidal marsh, Limnol. Oceanogr. Methods, 12(1 JAN), 10–22, doi:10.4319/lom.2014.12.10, 2014.

Rieger, L., Langergraber, G. and Siegrist, H.: Uncertainties of spectral in situ measurements in wastewater using different calibration approaches, Water Sci. Technol., 53(12), 187–197, doi:10.2166/wst.2006.421, 2006.

Shaw, B. D., Wei, J. B., Tuli, A., Campbell, J., Parikh, S. J., Dabach, S., Buelow, M. and Hopmans, J. W.: Analysis of ion and dissolved organic carbon interference on soil solution nitrate concentration measurements using ultraviolet absorption spectroscopy, Vadose Zo. J., 13(12), 1–9, doi:10.2136/vzj2014.06.0071, 2014.

---

## Author Comment (AC2) · 3 Dec 2020

General comments: This paper presents an advance in the analysis of soil water. Maybe not a giant leap but surely an interesting technique. The text is understandable and ïnË̆GCuent and the reader is able to get a fairly complete idea of the work done. However, some important points should be discussed.

We would like to thank the reviewer for the time and effort made in evaluating our paper, and we will do our utmost to explain the methods presented in this work, and answer the reviewer questions.

Comment 1: How easy is it for the farmer to prepare samples for analysis in a reproducible way?

Reply to general comment 1: It was not within our intentions that the farmer would be dealing with the dilution and spiking process described in the methods chapter. The purpose of this process is to obtain a matrix of samples with various DOC and nitrate concentrations which can be then used in the calibration process in a method for estimating nitrate concentration. This method can then be the analytical core for a LED-based nitrate sensor. Since this process requires previous knowhow and skills, the tasks of installation and calibration of such sensor would be done by a trained personal and not by the farmer. The only role of the farmer in such model is to send a porewater sample or soil samples to whom which providing the soil sensing facilities, and the sensor installation and calibration. Once the installation and calibration processes are completed the farmers will be able to use the calibrated probe for long durations without any further analysis. The calibration process was found to be robust enough to enable direct measurements. Yet, a site-specific it requires pre-calibration.

Comment 2: No explanation is given on how the estimation of ïn ËŻAgure 5 has been obtained.

Reply to general comment 2: Figure 5 presents predicted nitrate concentrations plotted against observed nitrate concentrations. The absorbance data used to calibrate the predicted nitrate concentration was measured at 235 nm by a standard laboratory spectrometer. Similar data is presented in figure 3, however, in this case, absorbance data were collected at 300 nm. Currently, 300 nm UV LED are readily available and can be used to perform nitrate analysis, as presented in this paper. However, the results presented in figure 5 demonstrates how a significant increase in the quality of

the nitrate analysis can be obtained when calibration is carried at 235 nm. As such, we believe that once 235 nm LED would become commercially available, it can greatly increase the accuracy of such sensor. We have revised the manuscript (lines 228-237) and figure 5 caption to make the addressed issue clearer.

Specific comments:

Comment 1: The authors state that "the polynomial calibration equation for ËĞ nitrate must be site-speciïn ËŻAc.". This can be a limitation to a system that should be ËĞ used by "farmers who are focused on food production in large scale agricultural setups".

Reply to comment 1: It is indeed correct that the calibration is a site-specific feature. However, since it is mainly affected by the composition of the DOC which is driven by the parameters such as soil type, climate conditions, fertilization methodology, source of organic matter, and other environmental aspects of the site. Therefore, the calibration made for a monitoring system that is installed under a field will continue to represent the site for a long duration. It may change only upon a dramatic change in the field conditions such as adding significant organic matter from a different source. In such a case the system may be recalibrated without the need to take it out of the soil. Moreover, preliminary research made when calibrating the system for a single point in a crop field shown that the calibration was accurate for measuring nitrate at porewater taken in other points of the crop field as well. This data is under preparation and we hope to publish it soon. Note that lengthy field observations of nitrate measurement by spectral analyses when reduction of DOC interference was done by a site-specific algorithm, showed that once calibration was obtained it remained valid for a prolonged time interval (2 years) (Yeshno et al., 2019). This implies that the chemical composition of the DOC and thus its spectral interference in the UV range fairly stable over time, and as such allow continuous reading of nitrate by spectral techniques.

Comment 2: Five soil samples can be few to represent all possible real situations.

Reply to comment 2: We agree with the reviewers comment; we indeed intend to

extend this research in order to map as many possible soils and investigate the connection between DOC type to compost type, soil type, and the DOC optical characteristics. However, this research was design to serve as a proof of concept, and as such, we chose six types of soil that would represent a large variety of agricultural soil types: (1) an open crop field with sandy loam soil located near the coastal plain (2) an inland open crop field with clay soil (3) a conventional vegetables greenhouse with sandy loam (4) an organic vegetables greenhouse with loamy soil (base on compost fertilization) (5) citrus orchards with loamy soil and (6) a general case of sandy loam mixed with commercial humus. A table was added to the manuscript (Table 1 – line 395) to present each study site location and soil type.

Comment 3: What is the accuracy of the value of Table 1?

Reply to comment 3: The average error value for the DOC and the nitrate is 0.1% and 6% respectively. The standard error values were added to the nitrate and DOC concentrations presented in table 2 (line 396), as well as to the data found in the supporting information file.

Comment 4: The authors state that "An important advantage of DOC ïnËĞCuorescence spectroscopy is that it is not affected by the presence of nitrate in ′ the solution." Sure? Why?

Reply to comment 4: Although nitrate and DOC have some similar absorption characteristics in the UV spectrum, only DOC has the chemical structural complexity which comprises the aromatic rings required to have fluorescence characteristics at the UV range. It is therefore that UV fluorescence spectroscopy is commonly applied for analyzing DOC in samples containing dissolved nitrate or iron instead of absorbance spectroscopy techniques (Bridgeman et al., 2011). The manuscript had been revised to clarify the addressed issue (lines 150 - 155)

Comment 5: It would have been better a version of Figure 3 where the nitrate prediction is based on the DOC measurement made with the ïnËĞCuorescence ′ technique.

Reply to comment 5: It is indeed possible to plot the estimated nitrate concentration in figure 4 (y-axis), by plugging in the predicted DOC concentrations in the nitrate site-specific polynomial equation. However, for technical issues, some of the data used to make the DOC calibration curves presented in figure 4 is not compatible with the data used to plot the observed vs predicted nitrate concentrations in figure 3. Still, the quality of correlation found between the fluorescence intensity to the DOC concentration is very high, with R2 no lower than 0.99 (Figure 4). Thus, predicted nitrate concentration calculated from DOC concentration obtained by fluorescence would be almost similar to predicted nitrate concentration calculated for known DOC concentration. We can however still exemplify this resemblance in the predicted nitrate concentrations for two cases: (1) citrus's orchard, and the (2) conventional greenhouse, where a sufficient amount of compatible DOC/nitrate data existed (Figure 1 at the current file).

Comment 6: Please explain why only the points of ïn ËŻAgure 3b are vertically aligned.

Reply to comment 6: We would like to deeply thank the reviewer for revealing a mistake in one of our data files. a thorough exam had exposed a typing mistake at an excel spreadsheet which had caused an error in the data plotted in figure 3b. We have corrected the figure and the supporting information file accordingly. Consequently, the data points in figure 3b should not normally appear vertically aligned.

Comment 7: Technical corrections the correct spelling is "humus" and not "hummus". Line 168, "Figure 3" is again "Figure 2".

Reply to Comment 7: comment accepted. The main manuscript, figures, figures caption, and supporting information files had been revised accordingly.

Comment 8: Line 172, "x" should be the multiplication sign

Reply to comment 8: comment accepted, the manuscript at Line 186 had been revised.

References:

Bridgeman, J., Bieroza, M. and Baker, A.: The application of fluorescence spectroscopy to organic matter characterisation in drinking water treatment, Rev. Environ. Sci. Biotechnol., 10(3), 277–290, doi:10.1007/s11157-011-9243-x, 2011.

Yeshno, E., Arnon, S. and Dahan, O.: Real-time monitoring of nitrate in soils as a key for optimization of agricultural productivity and prevention of groundwater pollution, , (2009), 3997–4010, 2019.

[Figure]

[Figure]

Figure 1 – Predicted vs. observed nitrate when predicted nitrate is estimated for: DOC values obtained by standard laboratory TOC analyzer (a&c) and DOC values obtained by fluorescence spectroscopy (b&d) for the case of the citrus orchard and the conventional greenhouse.

[Figure]

**Fig. 1.**

---

## Author Comment (AC3) · 3 Dec 2020

The Manuscript "A novel analytical approach for the simultaneous measurement of nitrate and DOC in soil water" by Elad Yeshno et al. presents a measurement setup which tests a combination of two analytical tools to demonstrate the potential of wavelength

specific detection of soil water nitrate concentrations in carbon rich soils. The Authors present in a very straightforward manner what they have done due to which reasons and how the outcomes support the choice of the combined measurement techniques presented in this paper.

The authors applied a combination of UV absorption spectrometry (around 300 nm) with fluorescence spectrometry (excitation/emission at 350/451 nm) to demonstrate the possible feasibility of this setup when applied with single banded LED technology. The study seems to be embedded in a research project or development program, which delivered already an interesting publication with regard to a possible application of the method (Yeshno et al. 2019, HESS (within the reference list this citation misses the journal. However, I have some points that need to be discussed prior to the editors decision about a possible publication in HESS.

Authors reply:

We would like to thank the reviewer for investing the time and effort needed to evaluate the work carried during this research. We however like to clarify that this research presents an analytical concept to estimate nitrate concentration in soil porewater containing Dissolved Organic Carbon (DOC). Furthermore, as this paper purely focuses on the conceptual level, it does not present a new measurement setup. As such, the results presented in the paper merely implies the possibility of using this concept as a ground for the development of a UV LED-based nitrate sensor.   Comments:

Comment 1: Generally, as a reader I would really appreciate the demonstration of a (-n inexpensive) LED-based technology which is able to directly measure in-situ soil water concentrations of DOC and nitrate. But it needs to be discussed whether a feasibility study for the application of analytical tools fits into the scope of HESS or better into another journal format (e.g. the EGU Journal "Geoscientific Instrumentation, Methods and Data Systems"). Reply to comment 1: Since the goal of the presented research was to provide a proof of concept for a new analytical approach, we have conducted the

experiments using standard laboratory equipment. However, we believe that the results presented in this study about the relation between DOC, nitrate and their absorbance characteristic can be the base for the development of an affordable, LED-based sensor. As such the future development may be of high interest for scientists from the fields of hydrology and environment.

Comment 2: The presentation of the study does not follow the typical/expected structure and misses some details: While the introduction section describes the background of the research problem and the linkage between DOC and Nitrate UV-absorption, it misses a review of other methods applicable to the problem presented in the paper. The interference of UV absorption of nitrate and DOC is known and several studies (some of them are cited in the paper, but in a different context) use two different wavelengths within the UV spectrum in combination with statistical models to overcome this problem. Even several commercial UV-Vis spectrometers for a parallel direct measurement of DOC and nitrate in aqueous solution are available on the market and widely in use e.g. in monitoring stations in wastewater treatment plants. The principle of this approach is at least similar to the one applied within this study and could be transferred to an application with LED-technology, likewise. The authors should explain how their approach improves the current state of the art technology (in their approach a second analytical tool is required).

Reply to comment 2: Indeed, there are few common and standard methods to deal with the interference to nitrate estimation caused by the UV absorption of DOC, however, to the best of our knowledge non were successfully tested for a range of concentrations of nitrate present in the porewater of agricultural root zone (Nitrate concentrations ranging from tens to thousands mg/L and DOC from tens to hundreds mg/L). For example, Ferree and Shannon, (2001) presented such method based on second derivative absorption spectroscopy for DOC in concentration up to 77 mg/L. However, this method is limited for N-nitrate concentration higher than 10 mg/L. An additional, similar method is carried by calibrating the nitrate concentration to reduction of twice the absorbance

intensity at 275 nm from the absorbance at 220 nm (Armstrong, 1963). However, this method can only be used when the absorbance at 275 nm is lower than 5 % of the absorbance measured at 220 nm. A further method that can be used to reduce interference from DOC is relying on a wide range absorbance measurement of the UV-VIS spectrum, combined with statistical tools, such as the Partial Least Square Regression (PLSR) (Avagyan et al., 2014; Etheridge et al., 2014; Rieger et al., 2006). Yet, a primary goal in this research was to develop a method that would serve as the basis for an affordable LED-based sensor for nitrate. Yet, the PLSR method, which requires UV-VIS absorption data at a broad spectrum cannot be used as a base for a narrow band, LED-based sensor. Lastly, none of the presented methods mentions the necessity for site-specific calibration. As shown previously in a research made by the author of this paper, due to the variability in the optical absorption characteristics of the DOC found at different agricultural sites, a site-specific calibration is required to perform adequate calibration for nitrate (Yeshno et al., 2019). The method described is based on preliminary sample chemical analyses of the DOC found in each site porewater, and obtaining a calibration equation that is dealing with the absorption/fluorescence characteristic found at each study site.

Comment 3: The description of the methodology regarding sampling and analytical procedures is very clear and to the point. The method development, e.g. the derivation of the 2D Model, which is then applied to the data, is presented within the result & discussion section. Here I would expect a clear difference between method description and acquired parameter values. As mentioned by reviewer 2, I would have expected a larger data set, covering a broader range of possible DOC concentrations than 6 and 25 ppm DOC, respectively.

Reply to comment 3: The comment is accepted and the part of the methods that is dealing with the sample preparation has been moved to the results section (lines 172-181). Regarding the presented data set, figure 1 merely presents an example leading the reader into the complexity of the superposition of the absorption spectroscopy caused

by the DOC and nitrate presence in the solution. We have calibrated and tested the concept for a broad DOC concentrations range, varying between 6 – 213 mg/L (table 1). Additional data of the DOC concentrations range tested in this research can be seen under section S2 in the supporting information file.

Comment 4: The specific results and the quality of the presented method are not presented in detail. In addition to the figures, there is only an overall description of the correlations between predicted and measured nitrate concentrations as well as the total range of RMSE. Since this is a new methodological setup, I would have expected at least a table, where the quality differences for the different field sites with regard to the differences in background DOC concentrations is presented to the reader.

Reply to comment 4: We accept the comment and a table of RMSE, R2, P-value, and DOC / Nitrate concentrations range had been added to the manuscript (table 4 line 402).

Comment 5: The discussion section does not compare the acquired results to other studies in the field (the authors present not a single citation of other literature from the field within this section).

Reply to comment 5: indeed, the discussion section does not compare the acquired results to other studies in the field, since it is not found necessarily in a comparable manner. As described in detail under comment 2, most of the standard commonly applied absorbance spectroscopy techniques are limited to 10 mg/L N-nitrate and to about 80 mg/L of DOC. These techniques are mostly suitable for laboratory analyses were dilutions of the obtained water samples can be made. Yet, since the current research focuses on porewater found in cultivated soils, we were expecting concentrations range higher in two orders of magnitude of nitrate and DOC. Additionally, these methods do not provide a site-specific solution to the local chemical and optical characteristics of the DOC. It is therefore that results obtained from the commonly used methods cannot be directly compared to the results found in this paper.

Comment 6: Since the presented method needs a site-specific calibration (comparable to the UV-vis-approach), the advantage of this method over others is not clear to me, e. g. whether this setup provides a higher accuracy than UV-vis based spectrometers. Overall, I recommend major revisions.

Reply to comment 6: The presented methods require site-specific calibration, to overcome the interference on the nitrate's absorbance spectrum, caused by the local DOC chemical composition in the soil. As shown in the previous work of Yeshno et al., (2019) the optical characteristics of DOC may differ from site to site, and as such compensation of interference from DOC cannot be associated only with its concentration and should account for its local chemical and optical characteristics as well. Most of the known standard, common methods to deal with the absorption caused by DOC (as detailly described under reply to comment 2 and comment 5) do not provide a site-specific solution for the local DOC, and as such was less suitable for in-situ, porewater analyses. Nevertheless, although statistical-based analyses such as PLSR and PLA shows potential for in-situ porewater analyses for nitrate, this method requires absorbance data on a broad spectrum of the UV-VIS light. As such, these methods are less suitable as an analytical core for the development of an LED-based sensor. Nevertheless, as discussed in the reply to comment 1 of reviewer #2, once the system is calibrated throughout the process of installation, the calibration curve may be relevant for long durations (years). Moreover, if needed the system can be recalibrated again (or from time to time) without the need to take the system out of the soil.

References:

Armstrong, F. A. J.: Determination of Nitrate in Water by Ultraviolet Spectrophotometry, Anal. Chem., 35(9), 1292–1294, doi:10.1021/ac60202a036, 1963.

Avagyan, A., Runkle, B. R. K. and Kutzbach, L.: Application of high-resolution spectral absorbance measurements to determine dissolved organic carbon concentration in remote areas, J. Hydrol., 517, 435–446, doi:10.1016/j.jhydrol.2014.05.060, 2014.

Etheridge, J. R., Birgand, F., Osborne, J. A., Osburn, C. L., Burchell, M. R. and Irving, J.: Using in situ ultraviolet-visual spectroscopy to measure nitrogen, carbon, phosphorus, and suspended solids concentrations at a high frequency in a brackish tidal marsh, Limnol. Oceanogr. Methods, 12(1 JAN), 10–22, doi:10.4319/lom.2014.12.10, 2014.

Ferree, M. A. and Shannon, R. D.: Evaluation of a second derivative UV/visible spectroscopy technique for nitrate and total nitrogen analysis of wastewater samples, Water Res., 35(1), 327–332, doi:10.1016/S0043-1354(00)00222-0, 2001.

Rieger, L., Langergraber, G. and Siegrist, H.: Uncertainties of spectral in situ measurements in wastewater using different calibration approaches, Water Sci. Technol., 53(12), 187–197, doi:10.2166/wst.2006.421, 2006.

Yeshno, E., Arnon, S. and Dahan, O.: Real-time monitoring of nitrate in soils as a key for optimization of agricultural productivity and prevention of groundwater pollution, , (2009), 3997–4010, 2019. Interactive comment on Hydrol. Earth Syst. Sci. Discuss., https://doi.org/10.5194/hess-2020- 417, 2020. C3